# A Case of Sarcoid-Lymphoma Syndrome with Various Etiological Factors

**DOI:** 10.3390/reports6020019

**Published:** 2023-04-23

**Authors:** Kazuki Furuyama, Makiko Tsukita, Yoichi Shirato, Yusaku Sasaki, Yugo Ashino, Toshio Hattori

**Affiliations:** 1Department of Respiratory Medicine, Sendai City Hospital, Sendai 982-8502, Japanyusaku@rm.med.tohoku.ac.jp (Y.S.); 2Department of Hematology, Sendai City Hospital, Sendai 982-8502, Japan; 3Research Institute of Health and Welfare, Kibi International University, Okayama 716-0018, Japan

**Keywords:** sarcoid-lymphoma syndrome, sarcoidosis, lymphoma

## Abstract

A 75-year-old female with a history of stomach cancer and depression was referred to our hospital for left cervical lymphadenopathy. The biopsy of her left cervical lymph node revealed noncaseating granulomas with multinucleated giant cells. The positron emission tomography/computed tomography (PET/CT) indicated general lymphadenopathy (left supraclavicular left axillary, hepatic lymph nodes), except for the hilar lymph node. Both histology by transbronchial lung biopsy (TBLB) and analysis of broncho alveolar lavage fluid (BALF) were indicative of sarcoidosis. C-reactive protein (CRP) and soluble interleukin-2 receptor (sIL-2R) were increased in the sera. An alternative cause of granulomatous disease was ruled out, and on follow-up, she was diagnosed with sarcoidosis. Two years later, she was hospitalized for fever, anorexia, lymph node tenderness, and erythema nodosum with significant increases in CRP and sIL-2R. After admission, the repetitive axillary lymph biopsy showed the same histological findings as before, but the G-band staining showed clonal abnormalities. Bone marrow biopsy revealed abnormal lymphocytes with petal-like nuclei. Finally, she was diagnosed with malignant lymphoma infiltrating the bone marrow. After CHOP-based chemotherapy, her laboratory data, lymphadenopathy, and clinical findings improved, and she was discharged from the hospital on the 90th day. Careful medical treatment, including genetic analysis of the lymph node, is necessary in patients with sarcoidosis if lymphadenopathy is predominant.

## 1. Introduction

Sarcoidosis is a multisystem disease characterized by noncaseating granuloma formation in any part of the body that affects young and middle-aged adults [1]. It mainly involves the skin, eyes, lungs, and heart, manifesting as erythema nodosum in the lower extremities, granulomatous uveitis, granular lung-field lesions with bilateral hilar lymphadenopathy, and atrioventricular block, respectively. For diagnosis of sarcoidosis, it is necessary to assess noncaseating epithelioid cell granulomas and exclude known causes of granulomatous reactions [1].

The aetiologia and pathogenesis of sarcoidosis remain undetermined. Coordination between antigen-presenting dendritic cells (DCs) and naïve CD4^+^ T-cells is important for the formation of granuloma in sarcoidosis [2]. The CD4^+^ T cells that trigger granuloma formation are strongly T_H_1-polarized. IFN*γ* and interleukin-12 (IL-12) are T_H_1-polarizing cytokines secreted by CD4^+^ T cells and DCs, respectively [3]. IL-18 is mainly produced by antigen-presenting cells, and IL-18 synergistically promotes the formation of sarcoid granulomas [4].

The formation of noncaseating granulation tissue is suggested to occur in response to endogenous or exogenous antigens [5] in genetically susceptible individuals and to induce an altered immune response leading to the disease. As exogenous antigens, fungi, viruses, or bacteria (Propionibacterium) cause immune system activation and are associated with sarcoidosis [6,7,8].

The formation of noncaseating granulomas can also occur through tumor antigens as with endogenous antigens [9]. This is known as tumor-related tissue. The sarcoid reactions result in the formation of epithelioid-cell granulomas. It is similar to a hypersensitivity reaction by antigenetic tumor factors [10,11]. Such sarcoid reactions may occur in lymph nodes draining an area housing a malignant tumor, the tumor itself, and even nonregional tissues. Overall, sarcoid reactions occur in 4.4% of carcinomas, in 13.8% of patients with Hodgkin’s lymphoma and in 7.3% of cases of non-Hodgkin lymphoma [12]. Sarcoid reactions are also caused by antigenic factors derived from the tumor cells.

Conversely, sarcoid-lymphoma syndrome is preceded by sarcoidosis followed by malignant disease. The development of lymphoma at least 1 to 2 years after the diagnosis of sarcoidosis was named sarcoid-lymphoma syndrome by Brincker [13]. The increased mitotic activity of lymphocytes in sarcoidosis due to the immune inflammatory response in the involved tissues has been postulated to increase the risk of the lymphocytes undergoing mutation and subsequent malignant transformation [14].

Those that cause persistent inflammatory irritation include CagY (the *Helicobacter pylori* (*H. pylori*) CagY Protein), and it predominantly drives interferon-gamma (IFN-γ) and interleukin-17 (IL-17) secretion by gastric CD4^+^ T cells in *H. pylori*-infected patients with low-grade gastric MALT lymphoma [15]. Although these reports have no clear causal relationship, there are a few reports regarding a possible association between *H. pylori* and sarcoidosis [16]. Therefore, the causes of sarcoidosis reaction may also be the cause of lymphoma.

Patients suffering from sarcoidosis were significantly more anxious and depressed than the general population [17]. In addition, there were statistically significant associations between depression and infection with Borna disease virus, herpes simplex virus-1, varicella zoster virus, Epstein–Barr virus (EBV), and Chlamydophila trachomatis [18]. These viruses may cause sarcoidosis [19]. Sarcoid-lymphoma syndrome is more common at an older age [12]. The late onset of sarcoid lymphoma syndrome may imply that different mechanisms are involved in its progression from those in other sarcoidosis. In other words, a stimulus induces a sarcoid-like reaction and, at the same time, causes lymphoma. However, it may take 2–3 years to assign the diagnosis, because the lymphoma changes are subtle.

In this study, we described a supposed sarcoid-lymphoma syndrome in which lymphoma developed 2 years after follow-up in an elderly patient with a diagnosis of sarcoidosis. The patient’s background included gastric cancer surgery, depression, EB virus, and *H. pylori* infection. A relationship between the onset of sarcoid-lymphoma syndrome and these diseases was explored.

## 2. Case Presentation Section

A seventy-five-years-old woman consulted the outpatient otorhinolaryngology department in our hospital because of left neck lymphadenopathy. She had noticed swelling in her neck several months prior to her visit, which had been increasing with time. A primary care doctor referred her to our hospital because magnetic resonance imaging (MRI) revealed multiple lymphadenopathies in her left neck. She was conscious, had no fever or pain, and no findings were noted in the oral cavity, nasopharynx, hypopharynx, or larynx. On palpation, several nodules 1.5 cm in size were detected in the left neck, left supraclavicular fossa, and left axilla. No skin, eye, muscle, or joint manifestations were observed, and no abnormalities were observed in the electrocardiogram.

She had a history of uterine myoma operation (at 45 years), gastric cancer (operation at 60 years), depression (at 65 years), and ileus (operation at 67 years).

The patient’s lab data showed no abnormalities except for an increase in soluble interleukin 2 receptor (sIL-2R) and a slight increase in C-reactive protein (CRP). Serum calcium, angiotensin converting enzyme (ACE), lysozyme, alkaline phosphatase (ALP), Serum protein electrophoresis and interferon r release assay (IGRA) for tuberculosis were negative (Table 1).

On chest CT and 18F-fluorodeoxyglucose (18F-FDG) PET, lymph node swelling was shown on the left neck, left supraclavicular fossa, left axilla, abdominal periaorta (Figure 1A,B), and hepatic portal region (not shown). However, there were no findings on the hilum of lung (Figure 1C). These lymph nodes demonstrated 18-FDG uptake in PET imaging (Figure 1D).

Histological examination was performed via biopsy of the left neck, left axilla, and lung tissue by transbronchial lung biopsy (TBLB). Hematoxilin and eosin (HE) staining showed destruction of the lymph node structure and tightly clustered epithelioid histiocytes, and occasionally multinucleated giant cells with few lymphocytes surrounded by fibrosis. Caseous necrosis or neoplastic cells were not observed (Figure 2). The BALF findings were consistent with sarcoidosis (macrophage: 71%; lymphocytes: 28%; neutrophils: 1.0%; eosinophils: 0%; basophils: 0%; and a CD4:CD8 ratio of 6.34.)

The pathological and clinical findings (cervical, supraclavicular fossa, left axillary supraclavicular fossa, and left axillary lymphadenopathy) were consistent with sarcoidosis. At this point, we diagnosed this case as sarcoidosis syndrome.

During the 2-year follow-up, she required seven hospitalizations for fever and leg pain. Until the fifth hospitalization, these symptoms were treated with rest and anti-inflammatory drugs. On the sixth hospitalization, oral prednisolone was started for the diagnosis of erythema nodosum in the lower extremities via examination skin biopsy. Prednisone was started at 30 mg and tapered to 5 mg. Prednisone administration continued for 10 months. However, the seventh hospitalization was due to fever, anorexia, and malaise, which were not controlled with steroids.

Her laboratory data indicated high WBC counts, and high levels of CRP and sIL-2R (Table 1). Endoscopic examination of the upper gastrointestinal tract revealed Barrett’s mucosa, so an *H. pylori* antibody test was performed, after which *H. Pylori* antibody was shown to be positive (12 µ/mL normal range < 10 µ/mL). PCR test for Epstein–Barr (EB) virus and anti-EB virus antibody were positive, suggesting chronic EB virus infection. On chest CT and 18F-FDG PET, in addition to multiple nodular shadows in the lung field, a new lymphadenopathy was observed in the mediastinum and left axilla.

The image of the chest CT and 18F-FDG PET (Figure 3) showed worsening of the sarcoidosis, but the patient did not respond to prednisone; therefore, malignant lymphoma was also considered, and a repeat axillary lymph node biopsy was performed. The histological findings were consistent with sarcoidosis.

Considering that she was deteriorating despite the administration of prednisone, that her sIL-2R was extremely high, and that she was elderly, G-band staining was performed on the new left axillary lymph node specimen. Metaphase images were observed in four cells via the G-banding method, and two of the four cells had chromosomal abnormalities that could be judged as clones, although the significance was unknown. Moreover, a bone marrow biopsy was performed. A very small number of abnormal lymphocytes with petal-like nuclei were detected and she was diagnosed with malignant lymphoma with bone marrow infiltration (Figure 4).

CHOP (Cyclophosphamide 400 mg/m^2^ Doxorubicin 25 mg/m^2^ Oncovin1 mg/body, Prednisolone 40 mg/m^2^) therapy was administered. CT imaging after treatment showed that the neck, axillary, hilar, and abdominal lymph nodes reduced in size. A blood test showed a marked improvement in inflammatory responses (CRP 0.24 mg/dL and sIL2R 1154 U/mL) and the patient was discharged on the 89th day after the last administration.

## 3. Discussion

Herein, we present an elderly female with sarcoid-lymphoma syndrome with a varied medical history. Initially, there were no clinical findings other than multiple lymphadenopathies. The pathological diagnosis of three sites was noncaseating epithelioid cell granulomas. Erythema nodosum was also observed during the course, which was consistent with sarcoidosis. During hospitalization with high fever, a fourth biopsy specimen revealed a chromosomal abnormality, and a bone marrow biopsy diagnosed lymphoma. Her medical history included gastric cancer surgery, depression, and EB virus and Helicobacter pylori infections. It is uncertain whether these factors caused sarcoidosis. Because a lymph node with noncaseating granulomas was detected, she was diagnosed with sarcoidosis. The diagnosis of sarcoidosis is not standardized but is based on three major criteria: a compatible clinical presentation, the finding of non-necrotizing granulomatous inflammation in one or more tissue samples (not always required), and the exclusion of alternative causes of granulomatous disease [20]. Her clinical symptoms were different from those of typical sarcoidosis in the following ways: (1) geriatric onset with multiple granulomatous lesions; (2) significant lymphadenopathy, predominantly on the left side. Moreover, lymphadenopathy in the hilar region was not observed, indicating that she was suffering from inflamed extrathoracic lymph nodes. Blood tests showed no findings characteristic of sarcoidosis except for elevated sIL-2R. Repeated lymph node biopsies, TBLB, and BALF led us to diagnose her with sarcoidosis. In addition, there was no evidence of infection, autoimmune diseases, or malignancy. She was placed under observation because her only clinical finding was lymphadenopathy.

However, 2 years later, she had to be hospitalized due to poor control of fever and extreme elevation of sIL-2R. The lack of steroid therapy responsiveness indicated that sarcoidosis was unlikely to be the cause of these symptoms, and we could not rule out the possibility that the patient had malignant lymphoma. For this reason, a bone marrow examination was performed. A very small number of atypical lymphocytes with petal-like nuclei were detected in the bone marrow, which allowed us to conclude that the patient had malignant lymphoma. Considering the progress after the diagnosis of sarcoidosis, it is highly likely that this was a case of sarcoidosis-lymphoma syndrome.

It is known that the onset of sarcoidosis precedes the diagnosis of the associated lymphoma by several years and the average is 10 years older than that of cancer-free patients with sarcoidosis [12]. The median age at sarcoidosis diagnosis is 48 years (range: 24–68 years), whereas the median age at lymphoma diagnosis is 63 years (range: 25–83 years) in France [21], which is similar to our case (75 y.o). On the contrary, the sarcoid-preceding type has an earlier age of sarcoid onset in the USA [22]. The patient suffered from gastric cancer and *H. pylori* infection. Helicobacter pylori causes persistent inflammation and induces MALT lymphoma [15] or gastric cancer [23]. In addition, sarcoidosis can be accompanied by *H. pylori* [16]. Long-term infection may be responsible for the development of the sarcoid response. To our knowledge, there were no reports of sarcoidosis-lymphoma syndromes with *H. pylori*-infection. After CHOP, this lymph node became smaller similar to the other lymph nodes.

In addition, the patient suffered from depression. Although many people with sarcoids present depressive symptoms [16], there is no report that demonstrates that depression causes sarcoidosis. However, there are reports suggesting a relationship between depressive symptoms and EB virus infection [18], and there is also a relationship between EB virus infection and sarcoidosis [20]. Thus, it is possible that depression and/or viral infection may have contributed to the development of sarcoidosis in this case.

Sarcoidosis is a systemic disease of unknown cause in which nonnecrotizing epithelial cell granulomas form in various organs throughout the body [24,25]. Epithelioid cell granulomas are found in a variety of infectious, allergic, and neoplastic diseases in addition to sarcoidosis. The presence of epithelioid cell granulomas without necrosis in the lymph nodes is characteristic of sarcoidosis but is not a specific finding. Therefore, a definitive diagnosis of sarcoidosis cannot be made based on histopathology alone but requires a comprehensive diagnosis in combination with clinical findings. The most important aspect is the exclusion of known causes of granulomatous reactions [1]. In the present case, elderly onset sarcoidosis was complicated by lymphoma. When a patient has a varied medical history, it is necessary to investigate whether these may lead to a malignancy, e.g., sarcoidosis may cause various malignancies. There are various reports on the relationship between sarcoidosis and malignancy [10,11,12,13,14].

There have been many reports of sarcoid-like reactions. However, in most cases, the diagnosis of malignant lymphoma is assigned based on the presence of typical cells on histological examination.

In the absence of such findings, a tissue biopsy from an organ other than the lymph node is required to detect abnormal lymphocytes. In the present case, the patient’s bone marrow biopsy revealed bone marrow infiltration of malignant lymphoma, and chemotherapy was successful in improving her state according to blood tests and imaging studies.

Substantial effort is required to distinguish between sarcoidosis as an autonomous disease and sarcoid-like reactions, because numerous other causes can give rise to this type of granulomatous infiltrate [26]. In order to solve these problems, a relationship between genetic factors and sarcoidosis was suggested, and it may be useful for differential diagnoses and exploring the genotype–phenotype relationship in sarcoidosis in the future [27,28]. When dealing with atypical sarcoid-like patients with varied medical histories, sarcoid-related malignant disease should be considered and genetic testing and bone marrow testing are also recommended.

## 4. Conclusions

We present a 75-year-old female patient with sarcoidosis-lymphoma syndrome who had been diagnosed with atypical sarcoidosis for 2 years. An increase in sIL-2R levels associated with indolent fever led us to perform an extensive investigation, including bone marrow biopsy. The histological examination of the bone marrow samples showed abnormal cells compatible with lymphoma infiltration. She received CHOP therapy and improved. Her medical history showed various etiological factors related with sarcoidosis. Careful treatment, including genetic analysis of the lymph node, is necessary in patients with sarcoidosis.

## Figures and Tables

**Figure 1 reports-06-00019-f001:**
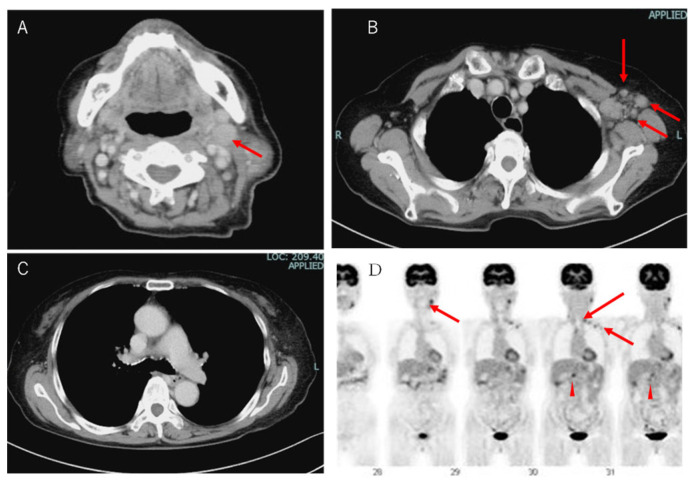
Image analysis of the patient. (**A**) Computed tomography (CT) images of the patient’s neck, (**B**) left axilla, (**C**) mediastinum and hilum of the lung. Left cervical lymph nodes and left axillary lymph nodes were swollen. Red arrows indicate swollen lymph nodes. (**D**) In the 18-FDG PET CT scan, FDG uptakes were seen in the left cervical lymph nodes, left supraclavicular fossa, and left axillary lymph nodes. In the abdomen, uptakes were indicated at the periaorta and hepatic portal region.

**Figure 2 reports-06-00019-f002:**
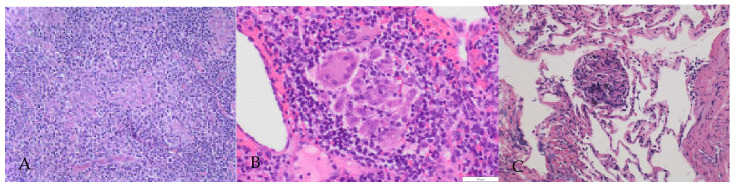
Histological findings of lymph nodes and lung. Numerous noncaseating granulomas. These granulomata exhibited a collection of epithelioid histiocytes with Langhans giant cells and were surrounded by fibrosis tissue and lymphocytes. (**A**) Left neck lymph node (HE × 50). (**B**) Left axilla (HE × 200). (**C**) Lung tissue (HE × 100).

**Figure 3 reports-06-00019-f003:**
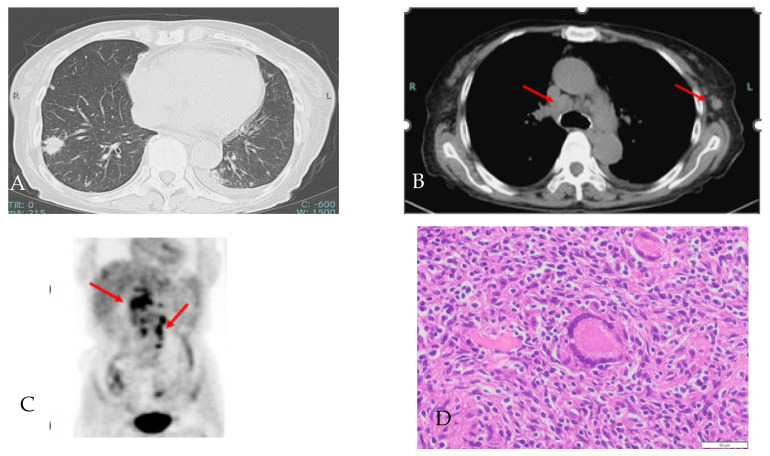
(**A**,**B**) Chest CT images of lung field and medical conditions, showing multiple nodule shadows in the bilateral lung, mediastinum lymph node enlargement (red arrow) and a new left axillary lymph node. (**C**) In the abdominal 18F-FDG PET, 18F-FDG uptakes showed an increased number of lymph nodes in the periaortic and hepatic portal region (red arrow). (**D**) Axillary lymph node histological findings showing numerous noncaseating granulomas with Langhans giant cells (HE × 100). Red arrows indicate swollen lymph nodes.

**Figure 4 reports-06-00019-f004:**
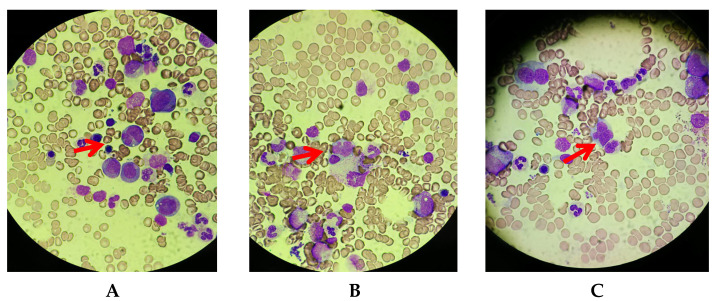
May–Grunwald–Giemsa staining of bone marrow (×400). A very small number of abnormal lymphocytes with petal-like nuclei were observed. (**A**–**C**) Separate fields of view. Red arrows are abnormal cells.

**Table 1 reports-06-00019-t001:** Laboratory data from the patient’s first visit and after 2 years.

Laboratory Data	Reference Range	First Visit	After 2 Years	Laboratory Data	Reference Range	First Visit	After 2 Years
Blood				Electrolyte			
Hematocrit (%)	42.0–53	39.1 *	34.1 *	Na (mmol/L)	138–145	140	136 *
Hemoglobin (g/dL)	13.5–17.5	12.2	10.5 *	K (mmol/L)	3.6–4.8	4.9	3.9
White-cell count (per mm^3^)	3700–8500	8100	27,000 *	Cl (mmol/L)	101–108	103	100 *
Differential (%)				Ca (mmol/L)	8.8–10.1	8.8	8.9
Neutrophils	44.0–68.0	67.5	90 *	P (mmol/L)	2.7–4.6	2.4	2.0 *
Bands	0.0–10.0	0	5.5	Coagulation system			
Metamyelocytes	0	0	0	PT (s)	10–13.5	11.0	12.6
Lymphocytes	27.0–44.0	24.7 *	2.5 *	PT-INR	0.8–1.2	0.96	1.1
Monocytes	3.0–12.0	4.4	1.5 *	ATPP (s)	23.0–38.0	31.3	51 *
Eosinophils	0.0–10.0	2.2	0.5	d-Dimer (µg/mL)	0.00–1.00	0.67	2.37 *
Basophils	0.0–3.0	1.2	0	Fibrinogen (mg/dL)	200–400	387	482 *
Platelet count (×10^3^ per mm^3^)	150–355	312	362 *				
Red-cell count (×10^6^ per mm^3^)	3.90–5.30	4.12	4.27				
Biochemical test				Urine			
Urea nitrogen (mg/dL)	2–80	12	22	Color	Yellow	Yellow	Yellow
Creatinine (mg/dL)	0.65–1.07	0.62 *	0.78	Clarity	Clear	Clear	Clear
ALT (U/L)	3–40	20	20	Specific gravity	1.009~1.025	1.036 *	1.016
AST (U/L)	8–35	17	32	pH	4.8~7.5	7.5	6.5
LDH (U/L)	124–222	143	211	Protein	-	-	-
ALP (U/L)	106–322	355 *	1102 *	sugar	-	-	-
Ferritin (ng/mL)	14–304	76	185	White cells per high-power field	-	-	4
C-reactive protein (mg/dL)	0.00–0.3	0.27 *	17.73 *	Red cells per high-power field	-	-	-
sIL-2R (U/mL)	122–496	1249 *	14,477 *	Virus Serological diagnosis			
ACE	7.0–25.0	14.9	23.0	EBV anti-EBNA (FA)	<10	20 *	10 *
lysozyme	5.0–10.2	6.9	14.5 *	EBV anti-VCA IgG (FA) EBV anti-EA-DRIgG (FA)	<10 <10	160 * <10	80 * <10
Total protein (g/dL)	6.6–8.4	6.9	5.2 *				
Albumin (g/dL)	3.8–5.2	4.0	2.7 *	HTLV-I anti-PA	<10	<10	<10
Serum proteins				EBV DNA Copy/mL			7 × 10^3^
A/G	1.55–2.55	1.71	1.33 *	IGRA (T spot)			
Albumin (%)	60.8–71.8%	63.1	57.0 *	NIL		0	
α1	1.7–2.9%	2.7	4.3 *	ESAT6		0	
α2	5.7–9.5%	7.6	9.0	CFP10		0	
β	7.2–11.1%	9.4	9.7	Positive control:		760	
γ	10.2–20.4%	17.2	20.0				

*: out-of-range values.

## Data Availability

The data used in this case report are available on reasonable request from the corresponding author.

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
