# Peer review of "A Case of Sarcoid-Lymphoma Syndrome with Various Etiological Factors"

_reports, 2023, doi:10.3390/reports6020019_

Round 1
Reviewer 1 Report
In the abstract try to underline why this case is unique and how it could change the approach in clinical practice.
KEY WORDS
Ok
The figure 1c does not show significant swelling lynphnodes; I suggest to choose a better slice
Crop CT images to enhance the visualization of lynphnodes
DISCUSSION
Summarize the clinical presentation. Try to compare to similar previous studies results.
Also try to describe the possible implications in clinical practice for example for clinicians dealing with sarcoidosis and sarco-related syndomes.
Reviewer 2 Report
This is an interesting case report about sarcoidosis-lymphoma syndrome, a combination of sarcoidosis and lymphoma. It is a rare but recognized clinical condition. Some manifestations may be common among these conditions, so differentiating each individual disease is challenging for the clinician, and even more so if they coexist in one same patient.
The authors described in detail a case of a 75-year-old female patient with sarcoidosis-lymphoma syndrome with atypical sarcoidosis. The patient's medical history includes surgery for gastric cancer, depression, and EB virus and Helicobacter pylori infections. However, it is unclear what factors caused sarcoidosis.
The authors have adequately described all clinical data and have accurately described sarcoidosis-lymphoma syndrome by inserting suitable references.
In my opinion, this case report is very interesting from a clinical point of view.
So, I therefore do not believe that changes to the text are necessary.
The various clinical data are appropriate to the purpose of the work and are accurately described.
The manuscript is generally well written and structured. The introduction provides sufficient background and include all relevant references.
I only suggest a few small changes:
1. in line 57 the word “lymphoma” is written in a different font from the rest of the text
2. add asterisks to the table next to out-of-range values
3. the sentence at line 193 is very similar to the sentence at line 170, find another way to explain the concept
4. at line 204 the word “nonnecrotizing” is written in a different font than the rest of the text
5. remove the yellow from line 228 “[25,26]”
6. remove the yellow from the word “references”
Also, the quality of the writing could have been much better.
However, I believe the article demonstrates high scientific value and is worth reading.
